# Genetic Variants of Glucose-6-Phosphate Dehydrogenase and Their Associated Enzyme Activity: A Systematic Review and Meta-Analysis

**DOI:** 10.3390/pathogens11091045

**Published:** 2022-09-14

**Authors:** Daniel A. Pfeffer, Ari Winasti Satyagraha, Arkasha Sadhewa, Mohammad Shafiul Alam, Germana Bancone, Yap Boum, Marcelo Brito, Liwang Cui, Zeshuai Deng, Gonzalo J. Domingo, Yongshu He, Wasif A. Khan, Mohammad Golam Kibria, Marcus Lacerda, Didier Menard, Wuelton Monteiro, Sampa Pal, Sunil Parikh, Arantxa Roca-Feltrer, Michelle Roh, Mahmoud M. Sirdah, Duoquan Wang, Qiuying Huang, Rosalind E. Howes, Ric N. Price, Benedikt Ley

**Affiliations:** 1Global and Tropical Health Division, Menzies School of Health Research and Charles Darwin University, Darwin 0810, Australia; 2Eijkman Research Center for Molecular Biology, Jakarta 10430, Indonesia; 3Infectious Diseases Division, International Centre for Diarrheal Disease Research, Bangladesh (icddr,b), Mohakhali, Dhaka 1212, Bangladesh; 4Shoklo Malaria Research Unit, Mahidol–Oxford Tropical Medicine Research Unit, Faculty of Tropical Medicine, Mahidol University, Mae Sot 63110, Thailand; 5Centre for Tropical Medicine & Global Health, Nuffield Department of Medicine, University of Oxford, Oxford OX1 2JD, UK; 6Médecins sans Frontières Epicentre, Mbarara Research Centre, Mbarara, Uganda; 7Mbarara University of Science and Technology, Mbarara 1956, Uganda; 8Fundaçāo de Medicina Tropical Dr. Heitor Vieira Dourado, Manaus 69040-000, AM, Brazil; 9Department of Internal Medicine, University of South Florida, Tampa, FL 33620, USA; 10Department of Cell Biology and Medical Genetics, Kunming Medical University, Kunming 650032, China; 11Diagnostics Program, PATH, Seattle, WA 98121, USA; 12Malaria Genetics and Resistance Unit, Institut Pasteur, INSERM U1201, 75015 Paris, France; 13Institute of Parasitology and Tropical Diseases, UR7292 Dynamics of Host-Pathogen Interactions, Federation of Translational Medicine, University of Strasbourg, 67081 Strasbourg, France; 14Yale School of Public Health, New Haven, CT 06520, USA; 15Malaria Consortium, Phnom Penh Center, Street Sothearos, Tonle Basac, Chamkarmorn, Building “H”, 1st Floor, Room No. 192, Phnom Penh, Cambodia; 16Malaria Elimination Initiative, Institute for Global Health Sciences, University of California, San Francisco, San Francisco, CA 94158, USA; 17Biology Department, Al Azhar University-Gaza, Gaza City, Palestine; 18Key Laboratory of Parasite and Vector Biology, Ministry of Health, National Institute of Parasitic Diseases, Chinese Centre for Disease Control and Prevention, Chinese Centre for Tropical Diseases Research, WHO Collaborating Centre for Tropical Diseases, National Centre for International Research on Tropical Diseases, Ministry of Science and Technology, Shanghai 200000, China; 19Chinese Center for Tropical Diseases Research, School of Global Health, School of Medicine, Shanghai Jiao Tong University, Shanghai 200240, China; 20School of Life Sciences, Xiamen University, Xiamen 361005, China; 21Foundation for Innovative New Diagnostics, 1202 Geneva, Switzerland; 22Mahidol-Oxford Tropical Medicine Research Unit, Faculty of Tropical Medicine, Mahidol University, Bangkok 10400, Thailand

**Keywords:** glucose-6-phosphate dehydrogenase, G6PD activity, G6PD deficiency, G6PD genotype

## Abstract

Low glucose-6-phosphate dehydrogenase enzyme (G6PD) activity is a key determinant of drug-induced haemolysis. More than 230 clinically relevant genetic variants have been described. We investigated the variation in G6PD activity within and between different genetic variants. In this systematic review, individual patient data from studies reporting G6PD activity measured by spectrophotometry and corresponding the G6PD genotype were pooled (PROSPERO: CRD42020207448). G6PD activity was converted into percent normal activity applying study-specific definitions of 100%. In total, 4320 individuals from 17 studies across 10 countries were included, where 1738 (40.2%) had one of the 24 confirmed G6PD mutations, and 61 observations (3.5%) were identified as outliers. The median activity of the hemi-/homozygotes with A-(c.202G>A/c.376A>G) was 29.0% (range: 1.7% to 76.6%), 10.2% (range: 0.0% to 32.5%) for Mahidol, 16.9% (range 3.3% to 21.3%) for Mediterranean, 9.0% (range: 2.9% to 23.2%) for Vanua Lava, and 7.5% (range: 0.0% to 18.3%) for Viangchan. The median activity in heterozygotes was 72.1% (range: 16.4% to 127.1%) for A-(c.202G>A/c.376A>G), 54.5% (range: 0.0% to 112.8%) for Mahidol, 37.9% (range: 20.7% to 80.5%) for Mediterranean, 53.8% (range: 10.9% to 82.5%) for Vanua Lava, and 52.3% (range: 4.8% to 78.6%) for Viangchan. A total of 99.5% of hemi/homozygotes with the Mahidol mutation and 100% of those with the Mediterranean, Vanua Lava, and Viangchan mutations had <30% activity. For A-(c.202G>A/c.376A>G), 55% of hemi/homozygotes had <30% activity. The G6PD activity for each variant spanned the current classification thresholds used to define clinically relevant categories of enzymatic deficiency.

## 1. Introduction

*Plasmodium vivax* has become the predominant cause of malaria outside of sub-Saharan Africa, causing between 4 and 14 million clinical cases annually [1,2]. The control and elimination of *P. vivax* is confounded by the parasite’s ability to form dormant liver stages (hypnozoites), which are not effectively eliminated by the schizontocidal drugs used to clear blood-stage infections [3]. Untreated *P. vivax* hypnozoites can reactivate weeks to months after the primary infection, causing recurrent episodes of malaria and ongoing transmission of the parasite [4]. The timely elimination of the parasite requires a radical cure, a combination of schizontocidal and hypnozoitocidal drugs, to kill the blood and liver stages of the parasite [5]. The only available class of drugs with hypnozoitocidal activity are the 8-aminoquinoline compounds (primaquine and tafenoquine), which cause severe haemolysis in individuals with glucose-6-phosphate dehydrogenase (G6PD) deficiency. G6PD deficiency (G6PDd) is a common inherited enzyme disorder, with a prevalence of 1% to 35% in malaria-endemic countries [6].

Exposure to several drugs and compounds can cause oxidative stress and induce haemolysis in G6PD-deficient individuals; these include 8-aminoquinoline agents, dapsone, ciprofloxacin, henna, and fava beans [7]. The risk of severe haemolysis following 8-aminoquinoline treatment is particularly relevant to the radical cure of patients with *P. vivax* malaria, and so the WHO recommends testing for G6PDd prior to administration of the antimalarial drugs [8]. The reference standard for diagnosing G6PD deficiency is quantitative UV spectrophotometry, for which several commercial kits are available [9,10]; however, spectrophotometry is not suitable for testing at the point of care [11,12]. In practice, routine diagnosis of G6PD deficiency is often unavailable or limited to more readily available and cheaper qualitative tests [13]. Tafenoquine is an 8-aminoquinoline drug that can be administered as a single dose; however, its recent licensing and roll-out requires quantitative G6PD testing prior to use, to identify patients with both intermediate (<70% normal activity) and severe (<30% normal activity) G6PD deficiency, in whom the drug is contraindicated [14]. Several new quantitative diagnostics have been developed to provide point-of-care testing for routine use [15,16].

G6PD deficiency is caused by one or more mutations in the *G6PD* gene, located on the X chromosome. Hence, males are hemizygous for the gene and phenotypically are either G6PD normal (G6PDn) or G6PDd, whereas females can be homozygous for the *G6PD* gene, conferring normal or deficient activity, or heterozygous, with activities that range from almost no activity to close to normal G6PD activities, with the majority clustering around the 50% activity threshold. A special case is compound heterozygous females who harbour two distinct *G6PD* variants on their two X chromosomes, both conferring low G6PD activities, similar to homozygous and hemizygous individuals. The G6PD gene was first cloned in 1986 [17], with subsequent studies identifying more than 230 mutations associated with reduced enzyme activity [7,18,19]. The majority of these arise from missense mutations, which cause substitution of a single amino acid [7]. Many G6PD mutations are rare, with limited observations from the few individuals reported in the literature. The large number of clinically relevant *G6PD* genotypes identified to date result in a wide range of phenotypes that are commonly characterised according to their residual G6PD enzymatic activity, but also according to other biochemical properties, such as electrophoretic mobility, thermal stability, and the Michaelis constant (Km). In 1971, Yoshida et al. proposed a classification of G6PD variants observed in hemizygous-mutated males according to five classes: I—severe enzyme deficiency, with chronic non-spherocytic anaemia; II—severe deficiency, with residual activity <10%; III—moderate-to-mild G6PD activity, with residual activity 10–60%; IV—very mild-to-no deficiency, with 60–100% residual activity; and V—increased G6PD activity, with >200% residual enzyme activity [20,21]. This classification system has been in use for the past 50 years and is incorrectly known as the “WHO classification”. To date the correlation between G6PD genotype and phenotype remains poorly characterized for many variants. In 2022, the World Health Organization Global Malaria Programme convened a technical consultation to propose a revised classification scheme for G6PD variants, spurred on by overlapping reports of G6PD activity in Classes II and III, and scarcity of reports in Class V [22].

To characterise the relationship between genotype and phenotype, and the associated variability, and to investigate its implications for classification of severity of G6PD variants, we undertook a systematic review and meta-analysis of the existing quantitative measurements of G6PD activity, in individuals with a known G6PD genotype.

## 2. Methods

MEDLINE (PubMed), Web of Science Core Collection (Clarivate), and SCOPUS were searched using standardized search terms (Appendix A; PROSPERO 2020 CRD42020207448). Studies were included for screening if they involved quantitative measurement of G6PD activity (using quantitative UV spectrophotometry at a wavelength of 340 nm) at a steady state (no haemolytic crisis within the previous 4 months) and molecular diagnosis of a G6PD variant known to be of clinical relevance. Each identified abstract was screened by at least two authors independently and a third author consulted for any disagreements (D.P., B.L., A.W.S., and A.S.). Full texts of relevant articles were then screened. Studies were excluded if they included only individuals with other known haematological conditions, newborns, or fewer than 20 G6PD normal males (with the exception of 2 studies reporting a robust definition of 100% G6PD activity), or if they did not provide sufficient information on laboratory procedures. Studies published before 2005 were excluded due to unavailability of individual-level datasets. The corresponding authors of the relevant studies were contacted at least twice and invited to provide published and unpublished individual patient data (IPD). Reference lists of the identified articles were screened for further relevant studies. Data confidentiality agreements were signed, and formal approval obtained, as required by the affiliated institutions of the corresponding authors.

The absolute values of the spectrophotometry results vary significantly between different laboratories [11]. Prior to pooling of G6PD activity observations from the different studies and settings, all measurements were converted from either U/g Hb or U/10^12^ red blood cell (RBC) to a % normal activity, using a study-specific definition of ‘normal’ (100%) G6PD activity. In most cases, this represented an adjusted male median (AMM) [9], either calculated from the included data (where datasets included ≥20 G6PD normal individuals) or using pre-defined values reported for each study. Whenever AMM or data from G6PD normal individuals were unavailable, an alternative definition of 100% activity was used, provided this was derived from the same study population in the same laboratory. To reflect the variability present in the genotyping methodology, individuals for whom no variant was confirmed were classed as either ‘wild-type’ (sequencing studies) or ‘no confirmed mutation’ (SNP-typing studies). Since it was not possible to discriminate between either scenario, IPD from these individuals were not analysed further. One study measured G6PD activity alongside 6-phosphogluconate dehydrogenase (6PGD) activity and reported the results as a ratio of the two enzyme’s activity. As the G6PD/6PGD ratio results exhibited a different dynamic range to G6PD activity alone, these data were not normalised to a fraction of normal activity, not included in the main analysis, and presented separately.

IPD were excluded if participants had a confirmed *Plasmodium* spp. infection, were less than one year of age, or zygosity was not defined. IPD were pooled and the median activity (in % of normal activity, or G6PD/6PGD ratio), interquartile range, and total range were calculated for each variant for homo-/hemizygotes and heterozygotes, separately. All variants for which data were available were included to indicate the breadth of the mutations present; however, since the available data for several variants was limited, variants were classed as data-rich (*n* ≥ 30 hemi-/homozygous deficient individuals) or data-poor (*n* < 30) [23]. G6PD activity estimates represent either a single point estimate (*n* = 1), a mean of two observations (*n* = 2), or the median of all G6PD spectrophotometry measurements (*n* ≥ 3). G6PD/6PGD ratio results were analysed separately, and excluded from analyses involving diagnostic thresholds (30%, 60%, 70%, and 80%) established for use with G6PD activity readings alone.

To highlight the presence of extreme measurements, which may reflect procedural errors, outliers were defined for data-rich variants. Outliers were defined per variant and separately for observations reported in U/gHb and those where the G6PD/6PGD ratio was defined. Any measurement that had an activity greater than 1.5× the interquartile range (IQR) above the median measurement for the respective variant (including all observations) was defined as an outlier [24]. Outlier measurements were retained, to illustrate the breadth of variability in measurements, but excluded from the estimates of G6PD activity, reported for each variant, and analyses involving clinical diagnostic thresholds. Differences in median readings were compared using the Kruskal–Wallis test with pairwise Wilcoxon post-tests, with Bonferroni correction to account for multiple comparisons. All analyses were performed using R version 4.0.3 [25].

To assess the risk of bias attributable to the study design and/or testing procedures, the QUADAS-2 tool [26] was modified (Appendix A) and applied to all studies. To assess whether a given study contributed a higher-than-average proportion of outlier measurements, the proportion of outliers in each study was compared to the overall dataset using chi-squared testing.

## 3. Results

### 3.1. Characteristics of the Pooled Database

A total of 838 papers were screened by title and abstract; 153 of these were included and the full text screened, and a further 10 papers were added from reference lists and author contact. Of these, 53 were identified as relevant (Figure 1); however, data from 36 papers were unavailable due to no author response or no permissions to share the IPD. Overall, data were available from 17 studies published between 2009 and 2021, conducted across 10 countries: 11 studies in Asia, 3 studies in the Americas, 2 studies in Africa, and 1 study in the Middle East (Figure 1, Appendix A) [27,28,29,30,31,32,33,34,35,36,37,38,39,40,41,42,43]. Individual-level data from the 4320 participants were available, of whom 564 (13%) individuals were excluded due to an age less than one year or unknown age, and 251 (6%) individuals aged above one year were excluded due to confirmed malaria infection, along with two (0.05%) females for whom it was unknown whether they were hetero- or homozygous [35,44]. Among the remaining 3503 individuals (81.1%), no clinically relevant G6PD variant was identified in 1765 (41%) individuals and these were therefore excluded from further analysis.

Overall, 1738 individuals (49.6%) had 1 of 24 clinically relevant *G6PD* variants, or A+, 1134 (65%) by SNP typing, and 604 (35%) by whole-gene sequencing (Table 1). In total, 556 (32%) individuals were hemizygous, 150 (9%) were homozygous, 20 (1%) were compound heterozygous, and 1012 (58%) were heterozygous. Spectrophotometric enzyme activity measurements were derived using spectrophotometry test kits from Trinity Biotech (Ireland) (*n* = 900, 52%), Randox (UK) (*n* = 299, 17%), Pointe Scientific (USA) (*n* = 21, 1%), or the WHO method (Beutler 1977) with WBC depletion (*n* = 276, 16%) [10,45,46]. A total of 242 individuals (14%) were included for which only the G6PD/6PGD ratio results were available (Zhongshan Biotech, China). A total of 61 readings were identified as outliers (Appendix A). There was no significant difference between the characteristics of individuals with readings identified as outliers and those with included readings.

### 3.2. Data-Rich Variants

A total of 7 variants had at least 30 observations in hemi-/homozygous individuals (Table 2). For variants analysed using % normal activity (*n* = 5), there was considerable variability in G6PD activity for both hemi-/homozygotes and heterozygotes (Figure 2 and Appendix A). After excluding outliers (*n* = 61), there were 67 hemi-/homozygotes with the A- (c.202G>A/c.376A>G) variant with a median G6PD activity of 29.0% (range: 1.7% to 76.6%). The corresponding median activities were 10.2% (range: 0.0% to 32.5%, *n*= 201) for the Mahidol variant, 16.9% (range: 3.3% to 21.3%, *n* = 45) for the Mediterranean variant, 9% (range: 2.9% to 23.2%, *n* = 36) for the Vanua Lava variant, and 7.5% (range 0.0% to 18.3%, *n* = 135) for the Viangchan variant. There was no significant difference in G6PD activity between the Mahidol, Vanua Lava, and Viangchan variants (p > 0.05), while G6PD activity for the A- (c.202G>A/c.376A>G) variant was significantly higher than all other data-rich variants (*p* < 0.001).

The variability in G6PD activity in heterozygous females was greater, ranging from 16.4% to 127.1% for A- (c.202G>A/c.376A>G) (median: 72.1%), 0% to 112% for Mahidol (median: 54.4%), 20.7% to 80.5% for Mediterranean (median: 37.9%), 10.9 to 82.5% for Vanua Lava (median: 53.8%), and 4.8% to 78.6% for Viangchan (median 52.3%).

Similar variability was observed for variants analysed as a ratio of the G6PD/6PGD activity. In total, 35 hemi-/homozygotes were included for both the Canton and Kaiping variants, with both variants exhibiting the same median G6PD/6PGD ratio of 0.4 (range: 0.1 to 0.7). The corresponding ratio was 0.3 (range: 0.1 to 0.8) for 25 individuals with the Gaohe variant (Table 3, Figure 3). Heterozygous individuals with these variants exhibited median G6PD/6PGD ratios of 1.5 (range 0.4 to 2.2, *n* = 36) for the Canton variant, 1.6 (range 0.5 to 2.0, *n* = 31) for Kaiping, and 1.1 (0.5 to 2.3, *n* = 24) for Gaohe.

### 3.3. Data-Poor Variants

A total of 15 variants had less than 30 observations in hemi-/homozygous individuals. Across these variants, G6PD activity followed the expected trends, with 60 hemi-/homozygous individuals falling below 30% activity. Overall, 14 data-poor variants had >1 observation, with similar levels of variability observed as for the data-rich variants. For example, for hemi/homozygous individuals with the Orissa variant, activity varied from 3.8% to 59.6%, with a similar spread for the hemi/homozygous individuals with Quing Yuan or Chinese-4 (20.0% to 46.4%). For eight compound heterozygous females, with one of seven combinations of G6PD mutations, all but one individual exhibited G6PD activity below 30%.

In 86 heterozygous females with a single, data-poor variant, enzyme activity ranged from 20% to 80% normal activity. Although not directly comparable to measurements expressed as a % normal activity, the same trends were observed for data-poor variants included using the G6PD/6PGD ratio method (Figure 4).

### 3.4. Diagnostic Implications of Observed Variability

Observations for data-rich variants were categorised according to the common diagnostic thresholds for either severe G6PD deficiency (<30% activity) or intermediate G6PD deficiency (<60%, <70% or <80%). For hemi/homozygous individuals, 99.5% of those with the Mahidol mutation and 100% those with the Mediterranean, Vanua lava, or Viangchan mutations had enzyme activity <30%. For A- (c.202G>A/c.376A>G), 55.2% of hemi/homozygotes had <30% activity in the six studies with A- observations. Overall, a 60% threshold included 25.7% A- (c.202G>A/c.376A>G) heterozygotes, 61.7% Mahidol heterozygotes, 71.4% Mediterranean, 65.0% Vanua Lava, and 68.1% Viangchan heterozygotes. These proportions increased to 59.3%, 88.5%, 85.7%, 99.3%, and 100.0%, respectively, when an 80% threshold was applied (Table 4).

### 3.5. Assessment of Study Quality and Risk of Bias

All included studies were assessed using a modified form of the QUADAS-2 tool (Appendix A) to examine the risk of bias towards the aims of this meta-analysis arising from the study design and/or sample collection and testing procedures [26]. The assessment was divided into four domains: patient selection, genotyping methods, spectrophotometry methods, and flow and timing. The included papers comprised a heterogenous mix of recruitment methods and study populations. Risk of bias due to patient selection was deemed high in seven studies, and unclear in two studies, due primarily to purposive selection of individuals with known G6PD deficiency, from a specific ethnic group or convenience sampling. A total of 13 out of 17 studies purposefully selected participants for genotyping based on prior phenotypic testing: only genotype-deficient individuals (or those less than 60% activity for example) or only a subset of G6PD normal individuals, resulting in a significant risk of bias towards the lower G6PD activity range. Genotyping methodology primarily introduced bias in the form of the selection of G6PD variants included in SNP-typing methods, providing logistical benefits but risked missing variants not included in the SNP-typing panel. While difficult to assess retrospectively, reported spectrophotometry methodologies were deemed appropriate in all included studies. The primary risk of bias identified in the spectrophotometry methodology was the absence of replicate measurements in 10 out of 17 included studies. This was deemed a high risk of bias due the documented potential for considerable inter-replicate variability in G6PD spectrophotometry [11]. Finally, when considering flow and timing, the primary source of bias identified was the exclusion of certain participants (e.g., G6PD normal individuals, etc.) from the final study samples, due to the wide range of study objectives and methodologies represented. All studies appeared to employ appropriate timing and storage of blood samples.

In addition to the above assessment, all included studies were assessed based on the proportion of outlier measurements that contributed to the data-rich variants. Two studies [31,39] (one comprising 50 A- individuals, the other contributing 22 Mahidol individuals) exhibited a significantly higher proportion of outlier measurements than other studies (*p* < 0.01). A sensitivity analysis was performed excluding all data from these studies, with little-to-no effect upon our overall findings (A- median G6PD activity 25.9% (range 1.7–64.4%), with 64% hemi/homozygous below 30% activity; Mahidol 10.2% (range 0.0% to 32.5%), with 99.5% hemi/homozygous below 30% activity.

## 4. Discussion

Our study highlights significant variation in G6PD activity for individuals with the same G6PD variant, which was apparently irrespective of the phenotypic method used. Whilst hemi-/homozygous-deficient individuals with the Mahidol, Mediterranean, Vanua Lava, and Viangchan variants had similar median enzyme activities (*p* > 0.05), their activities were significantly lower than activities of individuals with the A- (c.202G>A/c.376A>G) variant. Variation in G6PD activity was greatest for the A- variant (c.202G>A/c.376A>G), ranging from almost 0% to >100% across the six studies, even among hemi- and homozygous individuals. Enzyme activity varied least for the Mediterranean and Mahidol variants (ranging from 0% to 20% across three and five studies, respectively). For most variants, the observed G6PD activity distributions spanned the 10% threshold separating Class II from Class III in the 1971 classification of variant severity [21], supporting recent proposals for revised classes [22].

While considerable, this variability rarely resulted in a confirmed hemi-/homozygous individual crossing the 30% clinical threshold for severe deficiency. Individuals were categorised according to commonly used diagnostic thresholds at 30% and 70% G6PD activity. Almost all hemi-/homozygous individuals with the Mahidol, Mediterranean, Vanua Lava, and Viangchan variant were severely deficient (<30% activity, Table 4), with 29% of hemi-/homozygous individuals with data-poor variants (45/154) also falling below this line. At the same time, however, only 55% of hemi/homozygous individuals with the A- (c.202G>A/c.376A>G) variant had <30% activity and 3% had >70% activity and did not meet the criteria for being outliers. As expected, enzyme activity varied significantly in heterozygous females for all variants, ranging from very low activities to levels that would generally be categorised as normal, a reflection of lyonization [7].

Our findings highlight the substantial proportion of heterozygous individuals with activities between 60% to 80%, and this was apparent for all variants assessed. Out of all non-compound heterozygous females included, 29% had activities between 60% and 80%; hence, a relatively small change in assay precision or decision-making regarding treatment thresholds will have a large impact upon the number of individuals eligible for treatment. However, data on the haemolytic risk associated with G6PD activity in this range is limited, and this is likely to differ according to residual enzyme activity, the associated variant, and degree of oxidative stress.

Several factors may contribute to the observed variability in G6PD activity, including both laboratory and biological factors. Firstly, infancy is associated with elevated G6PD activity [47,48,49], and early reports suggest concurrent malaria infection may transiently increase G6PD activity [35,44]. To minimise the effect of these factors, malaria-positive individuals and infants below one year of age were excluded from our analysis. Second, although G6PD activities were normalised for each study, it is likely that some of the observed variability could be due to lab procedures or errors in data recording, demonstrated, for example, by the occurrence of two hemizygous males with the Mahidol variant that recorded a G6PD activity of more than 150% the normal, which were classified as outliers. Though unlikely, we cannot exclude the possibility that these individuals had Klinefelter syndrome, with an additional X chromosome [50]. Third, most assay protocols provided with commercial test kits do not require the removal of white blood cells from the sample prior to testing, despite extreme leucocytosis being known to influence G6PD spectrophotometry [51]. Since the leucocyte count was not done in most participants, this may have contributed to the observed variability. Fourth, two studies from 1975 and 2004 suggest potential diurnal variations in G6PD activity [52,53], although from relatively small sample sizes, and this may also contribute to the variation in observed G6PD activity. Finally, further variability may stem from sources such as unreported infection, recent haemolytic events, and undiagnosed haematological conditions, resulting in elevated reticulocytosis.

Our study has a number of limitations. First, retrospective analysis of spectrophotometry data cannot exclude poorly standardized and/or executed laboratory procedures [11]. To address this, all studies were assessed for quality control measures, and for the data-rich variants, extreme values were excluded. Additionally, the proportion of outliers contributed by each study was quantified and compared to the proportion of outliers in the complete dataset. Finally, a sensitivity analysis, excluding one dataset with a significantly higher proportion of outliers, did not alter the overall results significantly. This approach penalises small sample size studies, which may potentially introduce additional biases. Second, a total of 14 out of the 17 studies included in our analysis predominantly genotyped individuals below a pre-defined G6PD activity threshold. Accordingly, the derived G6PD activity distributions are likely to be skewed towards the lower end of the G6PD activity spectrum. This is particularly apparent when considering the activity range for heterozygous females with the Viangchan variant (Figure 2), where all observations were from studies in which only females with less than 60% or less than 80% activity were genotyped. Third, some variants were reported predominantly from a single study or from limited geographic areas. For example, almost 90% of all observations for the Mediterranean variant were reported by Reading et al. [37], and all observations of heterozygous females with the A- variant were derived from either the USA or Uganda [32,34,36,39]. Fourth, reported activities in U/gHb were normalized to percent activity to allow for a direct comparison of readings between studies, assuming equivalency between assays, which may not be the case for all assays included [10].

In conclusion, our results highlight marked variability in enzyme activity among individuals with the same G6PD variant. G6PD activity distributions spanned the widely used thresholds to demarcate classes for G6PD deficiency, supporting the updated classification schema recently proposed during a WHO-convened meeting of international experts [22]. In order to define the severity of deficiency associated with a given variant, genotyping should be performed, not only for individuals with a phenotypic activity below a pre-defined threshold, but also phenotypically normal individuals. Further studies are required to determine the association between the G6PD enzyme activity, genetic variant, and risk of severe haemolysis following an 8-aminoquinoline drug, to inform the diagnostic and clinical implications of this heterogeneity.

## Figures and Tables

**Figure 1 pathogens-11-01045-f001:**
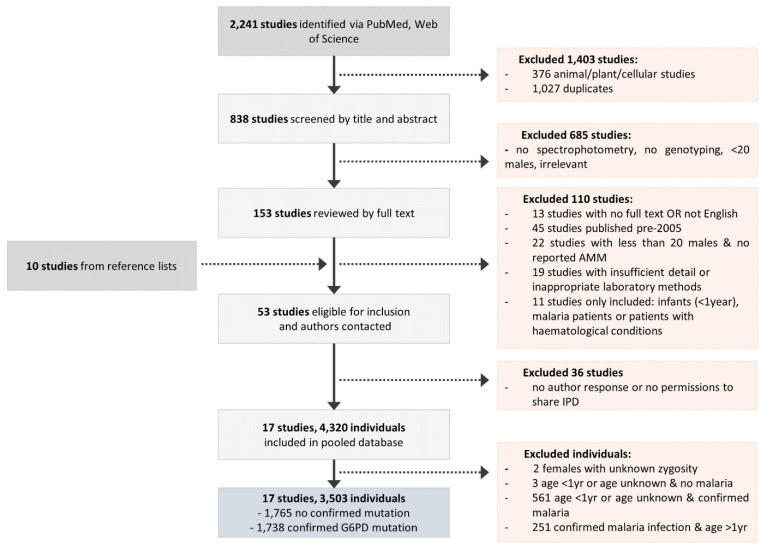
Flow diagram of the data collation procedure.

**Figure 2 pathogens-11-01045-f002:**
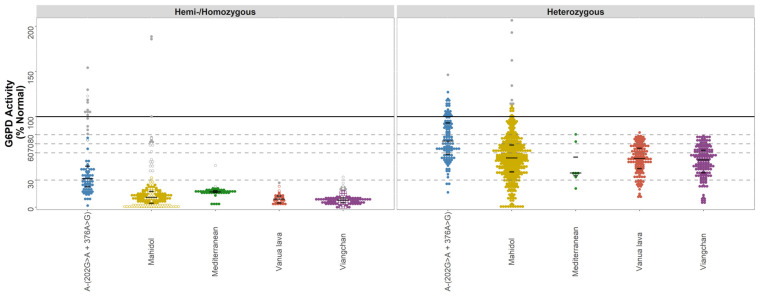
G6PD activity distributions (% Normal) for the data-rich variants—A-, Mahidol, Mediterranean, Vanua Lava, and Viangchan. Footnote: G6PD activity (% Normal) was measured by spectrophotometry for individuals with data-rich variants (*n* = 1296). The median and interquartile range of each variant are overlaid as black lines. Horizontal lines indicate diagnostic thresholds: 100% (black), 80%, 70%, 60%, and 30% (grey, dashed) G6PD activity. Homozygotes are indicated on the left panel using hollow points, and outliers are highlighted as grey points.

**Figure 3 pathogens-11-01045-f003:**
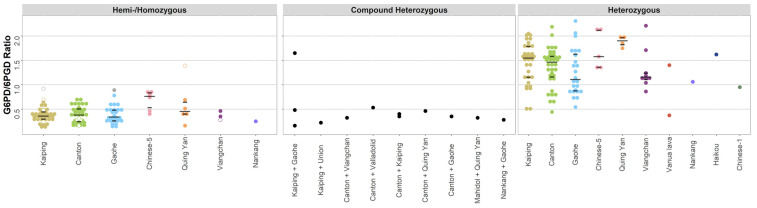
G6PD activity distributions for the variants investigated used the G6PD/6PGD ratio method. Footnote: G6PD activity (G6PD/6PGD) was measured by spectrophotometry (*n* = 242). The median and interquartile range of the variants with >3 observations are overlaid as black lines. Homozygotes are indicated on the left panel using hollow points, and outliers are highlighted as grey points.

**Figure 4 pathogens-11-01045-f004:**
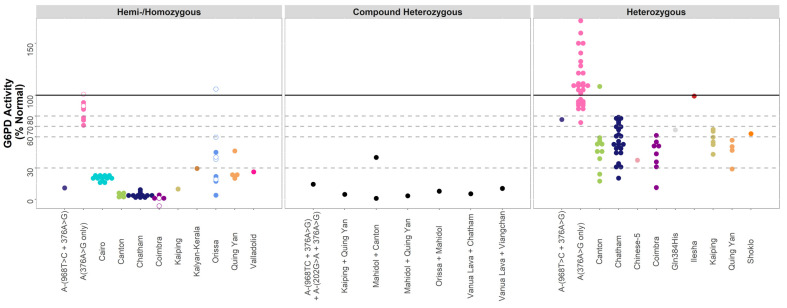
G6PD activity distributions (% Normal) for the data-poor variants. Footnote: G6PD activity (% Normal) was measured by spectrophotometry for individuals with data-poor variants (*n* = 154). The median and interquartile range of each variant are overlaid as black lines. Horizontal lines indicate diagnostic thresholds: 100% (black), 80%, 70%, 60%, and 30% (grey, dashed) G6PD activity. Homozygotes are indicated on the left panel using hollow points.

**Table 1 pathogens-11-01045-t001:** Characteristics of the individuals included in the pooled database (*n* = 1738).

	Hemi-/Homozygous *	Compound Heterozygous	Heterozygous *
**Sex**			
Female	150 (8.6)	20 (1.2)	1012 (58.2)
Male	556 (31.9)	-	-
**Age**			
Mean (range)	23.8 (1.0–78.0)	23.2 (1.0–40.0)	26.4 (1.0–75.0)
**Country**			
Bangladesh	94 (5.4)	-	50 (2.9)
Brazil	2 (0.1)	-	0 (0.0)
Cambodia	125 (7.2)	-	66 (3.8)
China	113 (6.5)	12 (0.7)	117 (6.7)
Indonesia	78 (4.5)	2 (0.1)	273 (15.7)
Myanmar	66 (3.8)	-	66 (3.8)
Palestine	64 (3.7)	-	4 (0.2)
Thailand	82 (4.7)	5 (0.3)	290 (16.7)
Uganda	58 (3.3)	-	79 (4.5)
USA	24 (1.4)	1 (0.1)	67 (3.9)
**Spectrophotometry Assay**			
Pointe Scientific	7 (0.4)	-	14 (0.8)
Randox	194 (11.2)	-	105 (6.0)
Trinity	352 (20.3)	4 (0.2)	544 (31.3)
WHO Method	40 (2.3)	4 (0.2)	232 (13.3)
Zhongshan Biotech	113 (6.5)	12 (0.7)	117 (6.7)
**Genotyping Assay**			
PCR	374 (21.5)	18 (1.0)	742 (42.7)
Sequencing	332 (19.1)	2 (0.1)	270 (15.5)
**Total**	706 (40.6)	20 (1.2)	1012 (58.2)

* Unless otherwise indicated, values represent *n* (%) of the total database in each category.

**Table 2 pathogens-11-01045-t002:** Median G6PD activity readings (% AMM) and variability of the different genetic variants.

	Variant Name	*n*	G6PD Activity Estimate (%)*	Interquartile Range	Range, (Min–Max)	Outliers (*n* (Min-Max))
**Data-rich Variants**	**A-(202A) ^c.376A>G + c.202G>A^**					
	Hemi-/Homozygous	67	29.0	20.1–38.1	1.7–76.6	14 (81.0–154.1)
	Heterozygous	113	72.1	58.0–93.2	16.4–127.1	1 (146.3)
	**Mahidol ^c.487G>A^**					
	Hemi-/Homozygous	201	10.2	3.0–15.3	0.0–32.5	22 (39.8–188.6)
	Heterozygous	381	54.4	38.8–68	0.0–112.8	7 (114.2–206.5)
	**Mediterranean ^c.563C>T^**					
	Hemi-/Homozygous	45	16.9	16.4–18.3	3.3–21.3	2 (23.0–46.0)
	Heterozygous	7	37.9	35–55.3	20.7–80.5	0.0
	**Vanua Lava ^c.383T>C^**					
	Hemi-/Homozygous	36	9.0	4.7–12.5	2.9–23.2	1 (27.0)
	Heterozygous	140	53.8	42.8–65.1	10.9–82.5	0.0
	**Viangchan ^c.871G>A^**					
	Hemi-/Homozygous	135	7.5	4.9–9.8	0–18.3	10 (19.2–33.3)
	Heterozygous *	160	52.3	38.2–63	4.8–78.6	0.0
**Data-poor** **Variants**	**A-(968C) ^c. 376A>G + c.968T>C^**					
	Hemi-/Homozygous	1	10.9	-	-	-
	Heterozygous	1	76.6	-	-	-
	**A(376G only) ^c.376GA>G^**					
	Hemi-/Homozygous	9	88.3	78.4–90.2	71.0–101	-
	Heterozygous	28	108.8	93.6–123.3	73.7–171.7	-
	**Cairo ^c.404A>C^**					
	Hemi-/Homozygous	12	20.2	19.0–22.4	15.5–23.4	-
	**Canton ^c.1376G>T^**					
	Hemi-/Homozygous	5	3.2	3.0–5.2	1.6–6.7	-
	Heterozygous	10	49.4	40.7–55.5	17.3–108.5	-
	**Chatham ^c.1003G>A^**					
	Hemi-/Homozygous	11	3.4	2.8–3.9	1.2–9.1	-
	Heterozygous	25	53.2	44.6–68.9	20.2–78.8	-
	**Chinese-5 ^c.1024C>T^**					
	Heterozygous	1	37.5	-	-	-
	**Coimbra ^c.592C>T^**					
	Hemi-/Homozygous	6	0.9	0.5–1.6	0.0–4.2	-
	Heterozygous	8	47.3	34.6–52.2	11.2–61.3	-
	**Gln384His^c.1152G>C^**					
	Heterozygous	1	66.7	-	-	-
	**Ilesha ^c.466G>A^**					
	Heterozygous	1	99.2	-	-	-
	**Kaiping ^c.1388G>A^**					
	Hemi-/Homozygous	1	9.7	-	-	-
	Heterozygous	6	57.3	53.1–63.8	43.2–67.2	-
	**Kalyan-Kerala ^c.949G>A^**					
	Hemi-/Homozygous	1	29.4	-	-	-
	**Orissa ^c.131C>G^**					
	Hemi-/Homozygous	9	38.3	19–45.1	3.8–105.9	-
	**Quing Yan or Chinese-4 ^c.392G>T^**					
	Hemi-/Homozygous	4	23.4	22.5–29.2	20.0–46.4	-
	Heterozygous	4	48.8	42.5–52.1	29.0–56.7	-
	**Shoklo ^c.701T>C^**					
	Heterozygous	1	63	-	-	-
	**Valladolid ^c.406C>T^**					
	Hemi-/Homozygous	1	26.2	-	-	-
	**A-(968C) + A-(202A)** ** ^c. 376A>G + c.968T>C + c.202G>A^ **					
	Compound Heterozygous	1	14.3	-	-	-
	**Kaiping^c.1388G>A^ + Quing Yan or Chinese-4 ^c.392G>T^**					
	Compound Heterozygous	1	4.7	-	-	-
	**Mahidol ^c.487G>A^ + Cantonc.^1376G>T^**					
	Compound Heterozygous	2	20.5	-	0.9–40.1	-
**Data-poor variants, Compound heterozygous**	**Mahidol^c.487G>A^ + Quing Yan^c.392G>T^**					
	Compound Heterozygous	1	3.3	-	-	-
	**Orissa^c.131C>G^ + Mahidol^c.487G>A^**					
	Compound Heterozygous	1	7.7	-	-	-
	**Vanua Lava ^c.383T>C^ + Chatham ^c.1003G>A^**					
	Compound Heterozygous	1	5.3	-	-	-
	**Vanua Lava ^c.383T>C^ + Viangchan ^c.871G>A^**					
	Compound Heterozygous	1	10.4	-	-	-

* Estimates for *n* = 1 are the single G6PD activity measurement; for *n* = 2, these are the mean of the 2 measurements; for *n* ≥ 3, these are the median of the included measurements. Estimates indicated in Columns 3–5 were calculated after the exclusion of outliers.

**Table 3 pathogens-11-01045-t003:** Median G6PD activity and variability of the variants investigated using the G6PD/6PGD ratio method.

Variant Name	*n*	G6PD Activity (G6PD/6PGD) *	Interquartile Range	Range, (Min–Max)	Outliers (*n* (Min–Max))
**Canton ^c.1376G>T^**					
Hemi-/Homozygous	35	0.4	0.2–0.5	0.1–0.7	-
Heterozygous	36	1.5	1.2–1.6	0.4–2.2	2 (9.4–9.5)
**Kaiping ^c.1388G>A^**					
Hemi-/Homozygous	35	0.4	0.3–0.4	0.1–0.7	1 (0.9)
Heterozygous	32	1.6	1.2–1.8	0.5–2	-
**Gaohe ^c.95A>G^**					
Hemi-/Homozygous	25	0.3	0.3–0.5	0.1–0.8	1 (0.9)
Heterozygous	24	1.1	0.9–1.6	0.5–2.3	-
**Chinese-1 ^c.835A>T^**					
Heterozygous	1	0.9	-	-	-
**Chinese-5 ^c.1024C>T^**					
Hemi-/Homozygous	6	0.8	0.5–0.8	0.4–0.8	-
Heterozygous	5	1.6	1.4–2.1	1.4–2.1	-
**Haikou ^c.835A>G^**					
Heterozygous	1	1.6	-	-	-
**Nankang ^c.517T>C^**					
Hemi-/Homozygous	1	0.2	-	-	-
Heterozygous	1	1.1	-	-	-
**Quing Yan or Chinese-4 ^c.392G>T^**					
Hemi-/Homozygous	6	0.5	0.4–0.6	0.2–1.4	-
Heterozygous	4	1.9	1.8–2.0	1.8–2.0	-
**Vanua Lava ^c.383T>C^**					
Heterozygous	2	0.9	0.6–1.1	0.4–1.4	-
**Viangchan ^c.871G>A^**					
Hemi-/Homozygous	3	0.3	0.3–0.4	0.3–0.5	-
Heterozygous	9	1.2	1.1–1.2	0.9–2.2	-
**Canton ^c.1376G>T^ + Gaohe ^c.95A>G^**					
Compound Heterozygous	1	0.3	-	-	-
**Canton ^c.1376G>T^ + Kaiping ^c.1388G>A^**					
Compound Heterozygous	2	0.4	0.4–0.4	0.3–0.4	-
**Canton ^c.1376G>T^ + Quing Yan or Chinese-4 ^c.392G>T^**					
Compound Heterozygous	1	0.5	-	-	-
**Canton ^c.1376G>T^ + Valladolid ^c.406C>T^**					
Compound Heterozygous	1	0.5	-	-	-
**Canton ^c.1376G>T^ + Viangchan ^c.871G>A^**					
Hemi-/Homozygous	1	0.3	-	-	-
**Kaiping ^c.1388G>A^ + Gaohe ^c.95A>G^**					
Compound Heterozygous	3	0.5	0.3–1.1	0.2–1.6	-
**Kaiping ^c.1388G>A^ + Union ^c.1360C>T^**					
Compound Heterozygous	1	0.2	-	-	-
**Mahidol ^c.487G>A^ + Quing Yan or Chinese-4 ^c.392G>T^**					
Compound Heterozygous	1	0.3	-	-	-
**Nankang ^c.517T>C^ + Gaohe ^c.95A>G^**					
Compound Heterozygous	1	0.3	-	-	-

* Estimates for *n* = 1 are the single G6PD activity measurement; for *n* = 2, these are the mean of the 2 measurements; for *n* ≥ 3, these are the median of included measurements. Estimates indicated in Columns 3–5 were calculated after the exclusion of outliers.

**Table 4 pathogens-11-01045-t004:** Number and percent of individuals falling into various diagnostic categories for data-rich variants.

Variant	*n* *	Studies (*n*)	Number (%) Included Using Diagnostic Thresholds *				
			<30%	<60%	<70%	<80%	≥80%
**A-(202A) ^c.376A>G + c.202G>A^**							
Hemi-/Homozygous	67	6	37 (55.2)	64 (95.5)	65 (97.0)	67 (100.0)	0 (0.0)
Heterozygous	113	4	3 (2.7)	29 (25.7)	52 (46.0)	67 (59.3)	46 (40.7)
**Mahidol ^c.487G>A^**							
Hemi-/Homozygous	201	5	200 (99.5)	201 (100.0)	201 (100.0)	201 (100.0)	0 (0.0)
Heterozygous	381	7	56 (14.7)	235 (61.7)	296 (77.7)	337 (88.5)	44 (11.5)
**Mediterranean ^c.563C>T^**							
Hemi-/Homozygous	45	3	45 (100.0)	45 (100.0)	45 (100.0)	45 (100.0)	0 (0.0)
Heterozygous	7	4	1 (14.3)	5 (71.4)	5 (71.4)	6 (85.7)	1 (14.3)
**Vanua Lava ^c.383T>C^**							
Hemi-/Homozygous	36	3	36 (100.0)	36 (100.0)	36 (100.0)	36 (100.0)	0 (0.0)
Heterozygous	140	3	18 (12.9)	91 (65.0)	119 (85.0)	139 (99.3)	1 (0.7)
**Viangchan ^c.871G>A^**							
Hemi-/Homozygous	135	6	135 (100.0)	135 (100.0)	135 (100.0)	135 (100.0)	0 (0.0)
Heterozygous	160	6	19 (11.9)	109 (68.1)	141 (88.1)	160 (100.0)	0 (0.0)

* Excludes outlier measurements.

## Data Availability

Data that contributed to this article can be obtained from the corresponding authors of each contributing study.

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
