# Peer review of "Genetic Variants of Glucose-6-Phosphate Dehydrogenase and Their Associated Enzyme Activity: A Systematic Review and Meta-Analysis"

_pathogens, 2022, doi:10.3390/pathogens11091045_

Round 1
Reviewer 1 Report
Pfeffer and colleagues report their results of a meta-analysis of studies linking G6PD genetic variants to enzyme activity. Improving our understanding of G6PD variants is critical, as 8-aminoquinoline drugs (e.g., primaquine and tafenoquine) used to treat P. vivax can induce hemolysis in patients with G6PD deficiency. There are additional reasons that improving our understanding of G6PD deficiency is warranted, given the gene's importance in food and drug interactions. I have very little in the way of suggestions for improvement of the work, as I found it compelling and well-described.
Major suggestions:
I think the paper would gain a broader reading if the authors were to expand the introduction on the importance of G6PD beyond 8-aminoquinoline drugs. Chloroquine and hydroxychloroquine are counter-indicated in G6PDd patients, for instance, and G6PDd-linked hemolytic anemia can be triggered by other drugs, foods, or infections.
The manuscript would benefit from a comparison of "normal" non-mutated G6PD levels as a sanity check before progressing to the analysis of variants. The same could be done with "data-rich" variants. Can you include (even as a supplement) a version of Fig. 2 split by study to confirm inter-study bias is minimized? Ideally this would be demonstrated statistically before moving ahead with the bulk of the manuscript.
Minor suggestions:
In title, "variant: should be "variants." But I would suggest the authors consider a pithier title that avoids "variability" and "variant". For suggestion: "Impact of genetic variants on G6PD enzyme activity: A systematic meta-analysis"
The opening sentence is a run-on sentence.
"Variants" does not always clearly refer to genetic polymorphisms. Change to: "We investigated the variation of G6PD activity within and between different *genetic* variants." (Line 48)
Insert comma after "In a systematic review" (Line 49)
I'm not sure what this refers to: "G6PD genotypes spanned current classification thresholds." Do you mean that the resultant enzyme levels associated with G6PD genotypes spanned current classification levels of G6PD deficiency severity? Please revise for clarity. (Line 62)
I would make the sentence beginning "To reduce the risk" to the start of the following paragraph. (Line 76)
Please note in the introduction that heterozygous females can also be symptomatic due to X-inactivation effects. (Line 95)
The "two hemizygous males with the Mahidol variant... with G6PD activity more than 150% of normal" could have Klinefelter syndrome (XXY). The odds are 1 in 500 to 1000, so it is not unexpected that there would be male individuals with two X-chromosomes in your sample of n=556.
Author Response
Reviewer #1
Pfeffer and colleagues report their results of a meta-analysis of studies linking G6PD genetic variants to enzyme activity. Improving our understanding of G6PD variants is critical, as 8-aminoquinoline drugs (e.g., primaquine and tafenoquine) used to treat P. vivax can induce hemolysis in patients with G6PD deficiency. There are additional reasons that improving our understanding of G6PD deficiency is warranted, given the gene's importance in food and drug interactions. I have very little in the way of suggestions for improvement of the work, as I found it compelling and well-described.
Major suggestions:
I think the paper would gain a broader reading if the authors were to expand the introduction on the importance of G6PD beyond 8-aminoquinoline drugs. Chloroquine and hydroxychloroquine are counter-indicated in G6PDd patients, for instance, and G6PDd-linked hemolytic anemia can be triggered by other drugs, foods, or infections.
We have revised the text and inserted the following sentence to the introduction to highlight the broader relevance of G6PD deficiency beyond the group of 8-aminoquinolines:
“Exposure to several drugs and compounds can cause oxidative stress and induce haemolysis in G6PD deficient individuals, these include 8 aminoquinolines agents, dapsone, ciprofloxacin, henna and fava beans[8]. The risk of severe haemolysis following 8-aminoquinolines is particularly relevant to the radical cure of patients with P. vivax malaria, and so the WHO recommends testing for G6PDd prior to administration of the antimalarial drugs.” (lines 88 – 92)
The manuscript would benefit from a comparison of "normal" non-mutated G6PD levels as a sanity check before progressing to the analysis of variants.
Although a respective comparison would be informative it is beyond the scope of the current analysis. In the studies included genotyping was targeted to identify the most common G6PD variants within each study area. Phenotypically deficient individuals with a variant that was not included in the genotyping attempt would be categorized as unknown and would not discriminate from G6PD normal individuals. Accordingly, we are unable to consider individuals where no known variant was identified.
We have clarified this in the methods: “To reflect the variability present in genotyping methodology, individuals in whom no variant was confirmed were classed as either ‘wild-type’ (sequencing studies) or ‘no confirmed mutation’ (SNP-typing studies). Since it was not possible to discriminate between either scenario, IPD from these individuals were not analysed further.“ (lines 156 – 159).
We have also clarified this in the results section, where it now reads: “Among the remaining 3,503 individuals (81.1%), no clinically relevant G6PD variant was identified in 1,765 (41%) individuals and these were therefore excluded from further analysis.” (lines 202 – 204)
The same could be done with "data-rich" variants. Can you include (even as a supplement) a version of Fig. 2 split by study to confirm inter-study bias is minimized? Ideally this would be demonstrated statistically before moving ahead with the bulk of the manuscript.
Many of the included studies contributed only 1 or 2 observations for a known variant, precluding direct statistical comparison. To clarify we have added a supplementary figure based on figure 2 to present the observed normalized activity (in %) per variant and study, that allows visual inspection of how results differ with study (S1 Figure, line 487)
Minor suggestions:
In title, "variant: should be "variants." But I would suggest the authors consider a pithier title that avoids "variability" and "variant". For suggestion: "Impact of genetic variants on G6PD enzyme activity: A systematic meta-analysis"
We have changed the title to “Genetic variants of Glucose-6-Phosphate Dehydrogenase and their associated enzyme activity: A systematic review and meta-analysis”
The opening sentence is a run-on sentence.
We have split the sentence in to two part. The beginning of the abstract now reads: “Low glucose-6-phosphate dehydrogenase enzyme (G6PD) activity is a key determinant of drug induced haemolysis. More than 230 clinically relevant genetic variants have been described.” (lines 56 – 58)
"Variants" does not always clearly refer to genetic polymorphisms. Change to: "We investigated the variation of G6PD activity within and between different *genetic* variants." (Line 48)
Added as suggested on line 59
Insert comma after "In a systematic review" (Line 49)
Added (line 59)
I'm not sure what this refers to: "G6PD genotypes spanned current classification thresholds." Do you mean that the resultant enzyme levels associated with G6PD genotypes spanned current classification levels of G6PD deficiency severity? Please revise for clarity. (Line 62)
We have revised the sentence which now reads: “G6PD activity for each variant spanned current classification thresholds used to define clinically relevant categories of enzymatic deficiency.” (lines 72 - 73)
I would make the sentence beginning "To reduce the risk" to the start of the following paragraph. (Line 76)
We have made the suggested change (lines 88 and 92)
Please note in the introduction that heterozygous females can also be symptomatic due to X-inactivation effects. (Line 95)
We have clarified this in the following sentence: “Hence males are hemizygous for the gene and phenotypically are either G6PD normal (G6PDn) or G6PDd, whereas females can be homozygous for the G6PD gene conferring normal or deficient activity or heterozygous with activities that range from almost no activity to close to normal G6PD activities, with the majority clustering around the 50% activity threshold.” (lines 103 – 106)
The "two hemizygous males with the Mahidol variant... with G6PD activity more than 150% of normal" could have Klinefelter syndrome (XXY). The odds are 1 in 500 to 1000, so it is not unexpected that there would be male individuals with two X-chromosomes in your sample of n=556.
This is an interesting comment. We have searched the literature on G6PD polymorphisms in individuals with XXY syndrome but cannot find any relevant publications. We observed two hemizygous males with the Mahidol variant with very high G6PD activities among a total of 201 hemizygous males with the Mahidol variant. Suggesting odds of 1:100, higher than what is described from the literature.
We have added the following sentence to the discussion: “Second, although G6PD activities were normalised for each study, it is likely that some of the observed variability is due to lab procedures or errors in data recording, demonstrated for example by the occurrence of two hemizygous males with the Mahidol variant recorded with G6PD activity more than 150% of normal that were classified as outliers. Though unlikely, we cannot exclude the possibility that these individuals had Klinefelter syndrome, with an additional X chromosome [50]).” (lines 364 – 369)
Reviewer 2 Report
The manuscript by Daniel A. Pfeffer et al. is of special interest in the field of malaria and particularly in the use of antimalarials to eliminate the disease. Although it is of particular importance to treat P. vivax carriers to avoid hypnozoite relapses and subsequent transmission to the community, however the high prevalence of G6PD deficiency in endemic areas is sufficiently worrying to indiscriminately treat with hypnozoite suppressants such as 8-aminoquinoline or primaquine without knowing their G6PD status, which is not always easy to assess in rural areas.
I would like to recognize the relevant and high amount of data provided in the manuscript since it is not easy to compile with such soundness. The careful methodological description demonstrates the great effort the authors have made to provide reliable and clean data, and I wish to congratulate them for this.
I have only found a few details which, if the authors could improve them, would surely help to provide some additional context for potential interested readers:
1- There is no bibliographic citation of the indicated 230 genetic variants of G6PD with clinical relevance (line 47). It would be relevant for the readers to provide a reference to the most recent paper compiling, a near complete, list of genetic variants, which I believe should be around 160. Within this information, it would be pertinent to specifically mention the number of those variants that are recognized as “polymorphic”, as these are precisely the ones that influence G6PD deficiency homogeneity in populations associated with malaria endemic areas. G6PD genetic homogeneity within populations is certainly related with the significance of the data provided in the manuscript.
2- It would be adequate to mention briefly in the abstract that the G6PD gene is X-linked so that the results provided in the groups hemizygosis/homozygosis and heterozygosis is understood.
3- I consider that a limitation of this study is the representation of the A- variant as it is only limited to Uganda and the United States, where Duffy-null may predominate. Given the importance of A- and its potential association or not with Duffy, perhaps it would be relevant to comment on this in the “limitations of the study” (discussion) so that the reader interested in this variant can be aware of its realistic representation.
4- I think this article would benefit from a brief sentence of general recommendation to clinicians, at the end of the discussion, for the geographical consideration where a variant is found. For example, the great variability found in G6PD A- activity suggest a recommendation of caution in its clinical assessment because of the connotations with anemias so frequent in Africa and with the existence of semi-immunity that provide a high prevalence of subclinical (hidden) malaria.
Author Response
Reviewer #2
The manuscript by Daniel A. Pfeffer et al. is of special interest in the field of malaria and particularly in the use of antimalarials to eliminate the disease. Although it is of particular importance to treat P. vivax carriers to avoid hypnozoite relapses and subsequent transmission to the community, however the high prevalence of G6PD deficiency in endemic areas is sufficiently worrying to indiscriminately treat with hypnozoite suppressants such as 8-aminoquinoline or primaquine without knowing their G6PD status, which is not always easy to assess in rural areas.
I would like to recognize the relevant and high amount of data provided in the manuscript since it is not easy to compile with such soundness. The careful methodological description demonstrates the great effort the authors have made to provide reliable and clean data, and I wish to congratulate them for this.
I have only found a few details which, if the authors could improve them, would surely help to provide some additional context for potential interested readers:
1- There is no bibliographic citation of the indicated 230 genetic variants of G6PD with clinical relevance (line 47). It would be relevant for the readers to provide a reference to the most recent paper compiling, a near complete, list of genetic variants, which I believe should be around 160. Within this information, it would be pertinent to specifically mention the number of those variants that are recognized as “polymorphic”, as these are precisely the ones that influence G6PD deficiency homogeneity in populations associated with malaria endemic areas. G6PD genetic homogeneity within populations is certainly related with the significance of the data provided in the manuscript.
We reference: Luzzatto L, Ally M, Notaro R: Glucose-6-Phosphate Dehydrogenase Deficiency. Blood 2020. (Reference #7). The suggested information is contained within this reference, specifically in tables 2, 3 and supplementary table 2.
2- It would be adequate to mention briefly in the abstract that the G6PD gene is X-linked so that the results provided in the groups hemizygosis/homozygosis and heterozygosis is understood.
Unfortunately, the word limit for the abstract is only 200 words. It is not possible to add further details to the abstract. We have however revised the introduction to make this clear:
“G6PD deficiency is caused by one or more mutations in the G6PD gene, located on the X-chromosome. Hence males are hemizygous for the gene and phenotypically are either G6PD normal (G6PDn) or G6PDd, whereas females can be homozygous for the G6PD gene conferring normal or deficient activity or heterozygous with activities that range from almost no activity to close to normal G6PD activities, with the majority clustering around the 50% activity threshold.’ (lines 102 – 106)
3- I consider that a limitation of this study is the representation of the A- variant as it is only limited to Uganda and the United States, where Duffy-null may predominate. Given the importance of A- and its potential association or not with Duffy, perhaps it would be relevant to comment on this in the “limitations of the study” (discussion) so that the reader interested in this variant can be aware of its realistic representation.
Data from hemi and homozygous deficient individuals with the A- variant were derived from 6 studies (Brito et al, Brazil; Johnson et al, Uganda; La Rue et al, USA; Pal et al, USA; Reading et al, Palestine; Roh et al, Uganda). As the reviewer points out data for heterozygous females with the A- variant were all either collected in the USA (La Rue et al and Pal et al) or Uganda (Johnson et al and Roh et al). To reflect this, we have added the following to the limitations section:
“Third, some variants were reported predominantly from a single study or from limited geographic areas. For example almost 90% of all observations for the Mediterranean variant were reported by Reading et al [37], and all observations of activities of heterozygous females with the A- variant were derived from either the USA or Uganda [32, 34, 36, 39].” (lines 390 – 395)
Describing the potential interaction of the Duffy factor and G6PD is beyond the scope of this review and is accordingly not included.
4- I think this article would benefit from a brief sentence of general recommendation to clinicians, at the end of the discussion, for the geographical consideration where a variant is found. For example, the great variability found in G6PD A- activity suggest a recommendation of caution in its clinical assessment because of the connotations with anemias so frequent in Africa and with the existence of semi-immunity that provide a high prevalence of subclinical (hidden) malaria.
Whilst we agree that our study has potential relevance for clinicians and policy makers, this would require a more nuance discussion and requires further consideration by advisory boards. This work is descriptive and does not assess the risk of hemolysis for specific agents which would be essential to understand the need for testing in a specific context. However, our analysis is already being considered by WHO to this effect. Hence, we would like to refrain from explicit clinical recommendations in the concluding remarks.